# Caveolae: Metabolic Platforms at the Crossroads of Health and Disease

**DOI:** 10.3390/ijms26072918

**Published:** 2025-03-24

**Authors:** Dante Maria Stea, Alessio D’Alessio

**Affiliations:** 1Facoltà di Medicina e Chirurgia, Università Cattolica del Sacro Cuore, 00168 Rome, Italy; dantemaria.stea01@icatt.it; 2Sezione di Istologia ed Embriologia, Dipartimento di Scienze della Vita e Sanità Pubblica, Università Cattolica del Sacro Cuore, 00168 Rome, Italy; 3Fondazione Policlinico Universitario “Agostino Gemelli”, IRCCS, 00168 Rome, Italy

**Keywords:** cardiovascular disease, caveolae, caveolins, endothelial cells, lipid rafts, metabolism, ocular disease

## Abstract

Caveolae are small flask-shaped invaginations of the plasma membrane enriched in cholesterol and sphingolipids. They play a critical role in various cellular processes, including signal transduction, endocytosis, and mechanotransduction. Caveolin proteins, specifically Cav-1, Cav-2, and Cav-3, in addition to their role as structural components of caveolae, have been found to regulate the activity of signaling molecules. A growing body of research has highlighted the pivotal role of caveolae and caveolins in maintaining cellular metabolic homeostasis. Indeed, studies have demonstrated that caveolins interact with the key components of insulin signaling, glucose uptake, and lipid metabolism, thereby influencing energy production and storage. The dysfunction of caveolae or the altered expression of caveolins has been associated with metabolic disorders, including obesity, type 2 diabetes, and ocular diseases. Remarkably, mutations in caveolin genes can disrupt cellular energy balance, promote oxidative stress, and exacerbate metabolic dysregulation. This review examines current research on the molecular mechanisms through which caveolae and caveolins regulate cellular metabolism, explores their involvement in the pathogenesis of metabolic disorders, and discusses potential therapeutic strategies targeting caveolin function and the stabilization of caveolae to restore metabolic homeostasis.

## 1. Introduction

The cell membrane, also known as the plasma membrane, is a dynamic, selectively permeable barrier that encloses the cell, maintaining structural integrity and mediating interactions with the extracellular environment. In the late 1950s, electron microscopy studies by Robertson suggested a trilaminar membrane structure, comprising two electron-dense outer layers attributed to proteins and a less dense central layer corresponding to the lipid bilayer [1]. However, subsequent research revealed that this model was overly simplistic. In the early 1970s, Singer and Nicolson introduced the fluid mosaic model, providing a more dynamic and functional perspective on membrane architecture [2]. According to this model, phospholipids are arranged with their hydrophilic heads facing outward toward the aqueous environment and their hydrophobic tails oriented inward, forming a nonpolar core that restricts passive diffusion of most substances. Integral membrane proteins are embedded within the lipid bilayer, often spanning its entire width, contrary to Robertson’s earlier suggestion that these proteins were merely surface-bound. These proteins are essential for maintaining plasma membrane functionality and structural integrity in eukaryotic cells, facilitating processes such as transport, signal transduction, cell recognition, adhesion, and interactions with extracellular matrix proteins to preserve membrane stability and cell shape [3]. Additionally, carbohydrate residues located exclusively on the extracellular surface can associate with membrane proteins to form glycoproteins or with lipids to produce glycolipids. The lipid bilayer also contains distinct microdomains, termed lipid rafts, with diameters ranging from 10 to 200 nm [4,5]. These microdomains are enriched in cholesterol and glycosphingolipids [5,6]. Their unique lipid composition and ability to cluster into larger domains play a crucial role in regulating the spatial distribution of molecules, coordinating cellular signaling and maintaining homeostasis [7,8,9,10]. However, defining the existence of lipid rafts has been challenging due to their small size, transient nature, and the limitations of conventional microscopy techniques in detecting them in living cells [11,12,13].

Caveolae, in contrast, are morphologically distinct flask-shaped invaginations of the plasma membrane, measuring 60–80 nm in diameter. These structures are primarily characterized by the presence of caveolin proteins, which are critical for their formation, structure, and function [14,15,16]. Although caveolae are more stable and structurally defined than lipid rafts, they can dynamically form endocytic vesicles [17]. This endocytic capability is particularly evident in the presence of viral agents; for example, Simian virus 40 has been shown to induce caveola-mediated trafficking into host cells within minutes of membrane binding [18]. Caveolae-mediated cholesterol-dependent endocytosis has also been reported for various viruses, including Hepatitis B virus, specific coronaviruses, and respiratory syncytial virus [19]. Furthermore, caveolae flattening into the plasma membrane is associated with cellular responses to mechanical stress [20,21]. It has been proposed that caveolae serve as a reservoir of plasma membrane that can be mobilized in response to mechanical stress or membrane damage [22]. This hypothesis is supported by in vitro findings demonstrating that RNA interference-mediated knockdown of caveolin-1 (Cav-1) leads to a substantial increase in EC size in vitro [23]. Beyond their structural role, caveolins regulate various cellular processes by binding and sequestering signaling proteins, including G protein-coupled receptors (GPCRs), receptor tyrosine kinases (RTKs), and Src family kinases. This regulatory function extends to cholesterol and sphingolipid homeostasis, endocytosis, transcytosis, cell signaling, and mechanosensing [24]. The essential role of caveolins in caveolae has been demonstrated in Cav-1-deficient mice, which completely lack caveolae [25]. However, caveolins also have functions independent of caveolae. In caveolae-deficient cells, such as hepatocytes and certain neurons, caveolins regulate lipid metabolism and intracellular trafficking [26,27], underscoring their functional versatility. Additionally, post-translational phosphorylation of Cav-1 at tyrosine 14 influences cellular processes such as signal transduction, vesicle trafficking, and cytoskeletal organization [28,29,30]. Notably, aberrant Cav-1 phosphorylation has been implicated in pathological conditions, including cancer metastasis and cardiovascular disease, highlighting its potential as a therapeutic target [31,32,33,34]. Despite decades of research aimed at distinguishing the structural and functional roles of lipid rafts and caveolae, significant ambiguities persist [11,12]. It remains challenging to delineate which cellular processes are specifically associated with lipid rafts, caveolae, or non-caveolar caveolins, underscoring the need for further investigation in this field.

## 2. Lipid Rafts, Caveolae, and Caveolins

Lipid rafts are specialized plasma membrane microdomains enriched in cholesterol, sphingolipids, and saturated fatty acids [4,5,35] (Table 1). Their structural organization is dictated by interactions between saturated lipids and cholesterol, which promote liquid-ordered phases that coexist with more fluid, unsaturated membrane regions. Cholesterol depletion, commonly induced using chemical agents such as methyl-β-cyclodextrin (MβCD) and Filipin, leads to rapid structural alterations, underscoring cholesterol’s critical role in raft stability and organization [10]. The “raft hypothesis” has been pivotal in redefining lipid functions, previously attributed primarily to membrane proteins. The lower fluidity of lipid rafts relative to surrounding membrane regions allows for them to serve as platforms for the assembly of signaling receptors [9,36,37]. Seminal studies have established lipid rafts as dynamic signaling hubs that regulate receptor clustering and activation, including pathways involving receptor tyrosine kinases and G protein-coupled receptors (GPCRs). The spatial organization of signaling molecules within lipid rafts facilitates intracellular signaling while restricting undesired interactions with non-raft-associated transducers, thereby modulating cellular responses. Notably, distinct raft subtypes may segregate signaling molecules in unstimulated cells while promoting their interaction upon specific stimuli [38]. Under conditions such as viral entry or epidermal growth factor receptor (EGFR) activation, lipid rafts have been observed to coalesce into larger and potentially more stable platforms that facilitate downstream signaling [38,39,40]. Although the precise roles of lipid rafts in cellular functions and the methodologies for their study remain active areas of research, current evidence suggests that these domains function as dynamic molecular platforms regulating membrane organization and signaling.

Caveolae, structurally distinct invaginations of the plasma membrane, are considered to be lipid raft subtypes based on their lipid composition and biophysical properties. First identified morphologically by Palade in the 1950s, caveolae exhibit unique structural and functional characteristics that differentiate them from other membrane microdomains [41]. Using electron microscopy, Palade observed plasma membrane invaginations in endothelial cells (ECs), termed “plasmalemmal vesicles”, suggesting their role in transcellular transport. Two years later, Yamada introduced the term “caveolae” (little caves) to describe 50–100 nm invaginations on gallbladder epithelial cells, characterized by the absence of a clathrin coat [42]. While research on clathrin-coated vesicles has advanced significantly, the biological and functional properties of caveolae remained elusive for decades due to the lack of definitive molecular markers. This challenge was overcome in the early 1990s with the identification of 17–24 kDa caveolins as key structural components of caveolae [43,44,45]. While research on clathrin-coated vesicles has advanced significantly, the biological and functional properties of caveolae remained elusive for decades due to the lack of definitive molecular markers. This challenge was overcome in the early 1990s with the identification of 17–24 kDa caveolins as key structural components of caveolae.

Methodologically, the unique lipid composition of lipid rafts and caveolae renders them resistant to solubilization by nonionic detergents, such as Triton X-100, at low temperatures [46]. This property has facilitated their isolation using detergent-based fractionation followed by density gradient ultracentrifugation [47,48,49,50]. Although variations in temperature and detergent concentration can influence experimental outcomes, this approach has been instrumental in characterizing the biochemical and functional properties of lipid rafts and caveolae across diverse physiological and pathological contexts. Collectively, caveolae and caveolins have emerged as key regulators of membrane organization and signal transduction, modulating GPCR signaling, Sarc-family kinases and nitric oxide synthase (NOS), with broad implications for cardiovascular diseases, metabolic disorders, and cancer biology.

### 2.1. Structural Features of the Caveolin Family of Proteins

Cav-1 is a 178-amino acid protein encoded by the CAV1 gene. It adopts an unconventional hairpin-like conformation within the membrane, with its hydrophobic domain embedded in the inner leaflet of the lipid bilayer [43]. Rather than spanning the membrane, this domain acts as an anchor, securing Cav-1 within the bilayer, while both its N-terminal and C-terminal domains remain cytoplasmic [51]. The N-terminal domain includes two key phosphorylation sites, Tyrosine 14 and Serine 80, which are essential for signaling and protein–protein interactions (Figure 1). This region also includes the highly conserved caveolin scaffolding domain (CSD), which facilitates interactions with key signaling molecules such as insulin receptors (IR) [52], endothelial nitric oxide synthase (eNOS) [53], and Src family tyrosine kinases [54]. The functional significance of CSD has been confirmed by mutational studies, demonstrating that alterations in specific residues disrupt interactions between Cav-1 and various signaling molecules [55,56,57], reinforcing its role as a central signaling hub [58]. CSD is believed to mediate interactions with both signaling and non-signaling proteins via the conserved caveolin-binding motif (CBM), a short hydrophobic sequence (8–11 amino acids) typically located on partner proteins [58]. However, the direct interaction between CSD and CBM remains a subject of debate due to the heterogeneous conformation of CBMs across different proteins and its limited accessibility to CSD [59]. In contrast, the C-terminal region of Cav-1 plays a crucial role in oligomerization and plasma membrane interactions [60]. Three palmitoylation sites within this region contribute to the recruitment of signaling proteins to caveolae, although they are not required for Cav-1 targeting to these domains [61].

The CAV1 gene encodes two major isoforms, α and β, generated through alternative splicing and exhibiting similar distribution patterns across cell types [62]. The α-isoform (24 kDa) consists of the full-length 178-amino acid sequence, whereas the β-isoform (21 kDa) lacks the N-terminal 32 amino acids, potentially affecting its membrane association and functional interactions. However, limited data exist regarding the distinct roles of these isoforms. Post-translational modifications, particularly phosphorylation, play a crucial role in regulating Cav-1 function. Tyrosine 14 is a primary phosphorylation site targeted by Src family kinases [63,64], while serine 80 phosphorylation has been implicated in cellular trafficking and apoptosis. Phosphorylated Cav-1 serves as a docking site for SH2 domain-containing proteins, facilitating the activation of downstream signaling pathways. Notably, Cav-1 phosphorylation has been linked to cancer progression, where it enhances tumor cell invasiveness by promoting focal adhesion turnover and cytoskeletal reorganization [30]. Emerging evidence suggests that Cav-1 not only contributes structurally to caveolae formation, but also plays a dual role in cancer progression, acting as a tumor suppressor in certain malignancies while promoting oncogenic signaling in others [31,65,66].

Cav-2, encoded by the CAV2 gene on chromosome 7q31.1, is a 20 kDa protein that exhibits the greatest sequence divergence among caveolin isoforms. Its specific role in mammalian cells remains less understood. Cav-1 and Cav-2 are predominantly expressed in endothelial cells, fibroblasts, pneumocytes, and adipocytes, where they form stable hetero-oligomeric complexes in vivo [67]. Similarly to Cav-1, Cav-2 contains two phosphorylation sites at specific serine residues, hypothesized to contribute to caveolae formation in coordination with Cav-1 [68]. Beyond its role as a molecular partner of Cav-1, Cav-2 has independent functions in pulmonary physiology. Studies have shown that Cav-2-deficiency develops pulmonary and endothelial dysfunction, highlighting its physiological importance [69,70].

Cav-3 is the muscle-specific isoform of the caveolin family, predominantly expressed in skeletal and cardiac muscle fibers [45]. It plays a key role in myoblast differentiation, myotube formation, T-tubule development, and muscle repair [71,72,73]. Cav-3 localizes to the sarcolemma of differentiated muscle fibers, but is absent in differentiating myoblasts [73]. Structurally, Cav-3 is a 22 kDa protein with a short N-terminal cytoplasmic domain, a single transmembrane domain and a long C-terminal cytoplasmic region essential for interactions with signaling molecules [74]. Analogous to Cav-1, Cav-3 regulates metabolic processes in skeletal muscle, including insulin signaling, glucose uptake, and nitric oxide synthesis [75,76,77]. It also plays a crucial role in cardiac mechanotransduction by modulating β_2_-adrenergic receptor (β_2_AR) signaling within caveolar microdomains. Mechanical stretch enhances β_2_AR-mediated cyclic adenosine monophosphate (cAMP) production and positive inotropy in cardiomyocytes, an effect dependent on intact caveolae. Notably, this response is abolished in Cav-3 knockout mice, underscoring its essential role in mechanosensitive signaling [78]. Mutations in the CAV3 gene have been linked to several muscle disorders, including autosomal dominant limb-girdle muscular dystrophy type 1C (LGMD-1C), characterized by progressive weakness of the pelvic and shoulder girdle muscles. Additionally, Cav-3 mutations are associated with rippling muscle disease, a condition marked by mechanically triggered muscle contractions [79,80].

### 2.2. Biogenesis of Lipid Rafts and Caveolae

Although lipid rafts and caveolae share a similar lipid composition, caveolae biogenesis is more complex, requiring specialized scaffolding proteins such as caveolins and caveolae-associated proteins to shape and stabilize these invaginations within the plasma membrane. While both structures function as lipid-rich ordered signaling hubs, they differ significantly in morphology, protein composition, stability, and cellular roles (Table 2). Caveolae formation and trafficking to the plasma membrane occur progressively, beginning with the synthesis of caveolin oligomers in the endoplasmic reticulum [81,82]. Cav-1 oligomers, consisting of 14–16 monomers, are transported to the Golgi complex, where they assemble into large, detergent-resistant complexes upon cholesterol binding. The crucial role of cholesterol in caveolae biogenesis is demonstrated by the complete loss of caveolae following treatment with cholesterol-depleting agents such as nystatin, filipin, or cyclodextrin (see Table 3 for a list of common chemicals affecting lipid rafts/caveolae biogenesis). These complexes are then directed to the plasma membrane, where they recruit accessory proteins, including Cavins, to facilitate the formation of characteristic caveolar invaginations [81,83,84]. Notably, cholesterol-depleting agents and lipid raft modulators, such as MβCD, filipin, and statins, are widely employed in experimental studies to investigate the functional roles of lipid rafts in cellular signaling, endocytosis, and disease pathogenesis. While these agents have provided valuable insights into membrane dynamics, concerns persist regarding their specificity and physiological relevance. Cholesterol depletion can compromise not only lipid raft integrity, but overall membrane structure, potentially disrupting non-raft-associated signaling pathways and leading to off-target effects. Furthermore, lipid raft modulators frequently trigger compensatory cellular responses, complicating data interpretation. Therefore, despite their utility in studying membrane microdomains, these tools should be used with caution. Experimental designs should account for their limitations and, whenever possible, be complemented by genetic approaches (e.g., caveolin knockdown models) to enhance specificity and minimize confounding effects.

Beyond caveolins, the Cavin protein family, including Cavin-1, Cavin-2, Cavin-3, and Cavin-4, is essential for caveolae biogenesis, acting as scaffolds that recruit Cav-1 to the membrane, a prerequisite for caveolar budding [109]. Proper Cavin protein localization is critical for regulating multiple signaling pathways. Notably, Cavin-2, in addition to its structural role, has been implicated in atherosclerosis progression in diabetic patients because of its modulation of eNOS activity and low-density lipoprotein (LDL) transcytosis, positioning it as a potential therapeutic target for cardiovascular disease [110].

Caveolae biogenesis and curvature regulation also involve accessory proteins such as Eps15 homology domain (EHD) proteins, which stabilize mature caveolae. The EHD protein family comprises ATP- and membrane-binding proteins, including EHD1, EHD2, EHD3, and EHD4. Among these, EHD2 is particularly critical for maintaining caveolae stability and curvature at the plasma membrane [111]. CRISPR/Cas9 knockout and knock in studies have demonstrated that EHD1 and EHD4 localize to the neck of caveolae and exhibit upregulated expression in the absence of EHD2, further underscoring their role in caveolae maintenance [112]. Additionally, members of the Pacsin protein family contribute to caveolae biogenesis, particularly by influencing membrane curvature. Three Pacsin isoforms have been identified: Pacsin-1, which is neuron-specific; Pacsin-2, which is ubiquitously expressed; and Pacsin-3, which is primarily found in muscle tissue. The structural characteristics and functional contributions of Pacsins to caveolae formation have been reviewed elsewhere [113]. Thus, caveolae biogenesis, structural integrity, and function depend on precise interactions among scaffolding proteins that distinguish these plasma membrane invaginations from lipid rafts, which lack such specialized components [114].

## 3. The Caveolar Network in Cell Metabolism and Metabolic Disorders

Metabolic disorders, characterized by disruptions in normal metabolic processes, represent a significant global public health challenge. Conditions such as obesity, type 2 diabetes mellitus (T2DM), dyslipidemia, and metabolic syndrome are strongly linked to cardiovascular diseases, non-alcoholic fatty liver disease, and other chronic illnesses [115]. Despite advances in elucidating the genetic, molecular, and environmental factors contributing to these disorders, their underlying molecular mechanisms remain incompletely understood. Emerging evidence highlights the involvement of caveolae and caveolin proteins in metabolic disorders, prompting extensive investigations into their molecular roles [116,117]. As discussed in this review, caveolins—particularly Cav-1 and Cav-3—are essential for maintaining caveolar structure and function and play key roles in insulin signaling, glucose uptake, and lipid metabolism [76,118]. Dysregulated caveolin expression has been implicated in metabolic disorders, including insulin resistance, T2DM, obesity, and cardiovascular disease (Table 4). These findings underscore the complexity of the caveolar network in metabolic homeostasis and suggests that it is a potential therapeutic target.

### 3.1. Caveolin Proteins in Glucose Metabolism and Glycolysis

Glucose metabolism is a fundamental pillar of cellular energy production, supporting a wide range of physiological processes essential for survival and function. Central to this pathway is glycolysis, a highly conserved multistep enzymatic cascade occurring in the cytosol of all cells. Glycolysis converts glucose into pyruvate, generating adenosine triphosphate (ATP) and reduced nicotinamide adenine dinucleotide (NADH) as primary energy intermediates [125]. Notably, glycolysis operates under both aerobic and anaerobic conditions. Under aerobic conditions, pyruvate is converted into acetyl-CoA, which enters the tricarboxylic acid (TCA) cycle and undergoes oxidative phosphorylation in mitochondria. Conversely, under anaerobic conditions, pyruvate is reduced to lactic acid, serving as the primary ATP source in tissues with low metabolic demands and limited mitochondria. Aerobic glycolysis is more energy-efficient, yielding a greater ATP output, albeit at a slower rate, whereas anaerobic glycolysis generates ATP more rapidly but with lower efficiency [78].

A distinctive feature of tumor cells is their reliance on aerobic glycolysis followed by lactic acid fermentation for energy production, even in the presence of sufficient oxygen—a phenomenon known as the Warburg effect [126,127]. This contrasts with normal cells, which primarily generate ATP through mitochondrial oxidative phosphorylation under aerobic conditions. Warburg initially hypothesized that this metabolic shift in cancer cells stemmed from impaired oxidative phosphorylation due to mitochondrial dysfunction [128]. However, other authors refuted this, demonstrating that oxidative phosphorylation rates in tumor cells were comparable to those in normal cells, suggesting intact mitochondrial function [129]. A plausible explanation for this metabolic preference is that glycolysis provides ATP at a faster rate, conferring a selective advantage to rapidly proliferating tumors. Supporting this hypothesis, cancer cells frequently overexpress glucose transporters (e.g., GLUT1) and key glycolytic enzymes, enhancing glucose uptake and metabolism [130]. The Warburg effect underlies FDG-PET imaging, where fluorodeoxyglucose-18 (FDG), a radioactive glucose analog, detects tumors due to their elevated glucose uptake. Advances in this field are expected to yield novel cancer therapies targeting tumor metabolic vulnerabilities [131,132,133].

Recent discoveries in cell biology highlight that glycolysis regulation extends beyond classical enzymatic mechanisms, implicating caveolae and their associated caveolin proteins in metabolic modulation. In cancer, several studies have linked Cav-1 to metabolic reprogramming. Tahir et al. demonstrated that Cav-1 interacts with low-density lipoprotein receptor-related protein 6 (LRP6), forming a signaling platform that activates insulin-like growth factor receptor (IGF-IR) and IR, subsequently stimulating the Akt/mTORC1 pathway and promoting aerobic glycolysis in prostate cancer cells [134]. Similarly, Díaz-Valdivia reported that phosphorylated Cav-1 facilitates the metabolic shift by enhancing glycolysis while inhibiting mitochondrial respiration, leading to increased reactive oxygen species (ROS) levels, which further drive Cav-1-induced cell migration and invasion in melanoma cells [135]. Ha and colleagues showed that Cav-1-mediated aerobic glycolysis in colorectal tumor cells is promoted by high-mobility group A1 (HMGA1)-mediated transcription of the SLC2A3 gene, encoding a glucose transporter, thereby increasing glycolytic flux [136]. In pancreatic cancer, Cav-1 has been associated with heightened glycolysis, contributing to cancer-induced cachexia, a syndrome characterized by severe weight loss and muscle wasting. Cav-1 interacts with IGF1R and IR, underscoring its role in metabolic alterations that support tumor growth and progression [137]. These findings highlight the multifaceted role of Cav-1 in glycolysis modulation within tumor cells, reinforcing its involvement in metabolic reprogramming that facilitates cancer progression.

Mutations in the CAV-3 gene have been implicated in compromised plasma membrane integrity and alterations in essential muscle cell processes, leading to skeletal muscle disorders that are collectively termed “caveolinopathies” [138]. Similarly to Cav-1, Cav-3 has been identified as a regulator of glucose metabolism in C2C12 muscle cells overexpressing Cav-3. These cells exhibit increased membrane localization of Cav-3 and GLUT4, enhancing glucose uptake and cell proliferation via Akt signaling pathway activation [77]. Furthermore, Cav-3 has been shown to associate with phosphofructokinase-M, a key glycolytic enzyme, in a glucose-dependent manner, supporting its role in metabolic regulation and its potential role in skeletal muscle pathophysiology [139].

### 3.2. Caveolae and Caveolin Proteins in Insulin Signaling and Diabetes

Following a meal, carbohydrates are digested into glucose, leading to an increase in blood glucose levels. In response, pancreatic beta cells within the islets of Langerhans secrete insulin, which rapidly induces autophosphorylation and activation of the IR, a member of the receptor tyrosine kinase superfamily [140]. Insulin facilitates glucose uptake via GLUT4 transporters in muscle and adipose tissue, promotes glycogen synthesis in the liver and muscle, and suppresses hepatic gluconeogenesis, thereby reducing blood glucose levels.

Immunolabeling and biochemical analyses in adipocytes have demonstrated that insulin stimulates GLUT4 and caveolin translocation from intracellular stores to caveolae, enhancing glucose uptake. Sucrose gradient fractionation of these cells revealed that GLUT4 translocation to the plasma membrane corresponded with a decrease in its concentration in heavy gradient fractions containing most cellular membranes [141,142]. A similar mechanism has been proposed in muscle cells, where GLUT4 localizes to plasma membrane invaginations known as transverse (T-) tubules, which likely form through multiple caveolar budding events [143]. The role of the caveolar network in insulin signaling is further supported by studies showing that caveolae disruption—via cholesterol-extracting agents such as β-cyclodextrin or expression of a dominant-negative Cav-3 variant in muscle cells—significantly impairs insulin-induced GLUT4 translocation to the plasma membrane [144,145]. Additional investigations using small interfering RNA targeting Cav-1 have indicated that, while caveolae are not essential for GLUT4 translocation or insulin-stimulated glucose uptake, they are critical for GLUT4 internalization following insulin withdrawal, highlighting their role in post-translocation processes rather than initial uptake [146]. Notably, IR is specifically localized within caveolae, where it directly interacts with Cav-1 via its scaffolding domain, modulating insulin signaling pathways [52]. Subsequent studies have identified a distinct caveolin-binding motif within the cytoplasmic tyrosine kinase domain of IR, which mediates interactions with Cav-1 and Cav-3 [147]. While early studies did not detect the insulin-induced phosphorylation of caveolin [142], later research demonstrated that insulin triggers a dose-dependent phosphorylation of Cav-1 at Tyr14 via IR in 3T3-L1 adipocytes. Although the precise molecular mechanisms underlying this process remain unclear, evidence suggests a potential link to adipocyte differentiation, as it is absent in preadipocytes [148,149,150]. Cav-1 also plays a crucial role in IR internalization and recycling, thereby modulating insulin signaling efficiency. The literature supports its function in stabilizing IR at the plasma membrane and facilitating its phosphorylation. Moreover, Cav-1 and the small GTPase Cell Division Cycle 42 (Cdc42) are key regulators of cytoskeletal remodeling and vesicle trafficking, ensuring insulin exocytosis in pancreatic beta cells [151]. Under basal conditions, Cav-1 binds inactive, GDP-bound Cdc42; glucose stimulation promotes Cav-1 dissociation, activating Cdc42 and ensuring the precise timing and localization of insulin secretion. This regulatory network is further supported by in vivo studies demonstrating that Cav-1 knockout mice exhibit pronounced insulin resistance, whereas Cav-1 upregulation enhances insulin signaling and insulin-induced glucose uptake [119,152,153]. Thus, disruptions in caveolar integrity or abnormal caveolin expression are closely linked to key diabetic pathophysiologies, including insulin resistance, beta-cell dysfunction, and vascular complications [120,151,154]. These findings expand our understanding of Cav-1’s role in metabolic regulation across tissues, highlighting its potential as a therapeutic target in diabetes. Enhancing Cav-1 expression could improve IR localization and stabilization within caveolae, thereby increasing insulin sensitivity. Alternatively, Cav-1 mimetic molecules may enhance IR signaling in insulin-resistant tissues. While these strategies remain speculative, further elucidation of Cav-1’s role in insulin signaling may pave the way for novel therapeutic approaches to diabetes and related metabolic disorders.

### 3.3. Caveolae and Caveolin Proteins in Obesity

Obesity is a major global health challenge, closely associated with metabolic disorders such as insulin resistance, type 2 diabetes, and cardiovascular disease. Extensive research has highlighted the pivotal role of caveolae and their integral membrane proteins, caveolins, in lipid metabolism and energy homeostasis. In adipocytes, Cav-1 is involved in key cellular processes, including lipid storage [155], fatty acid uptake [156], and insulin signaling [52]. Notably, among cell types characterized by a high abundance of caveolae and caveolin proteins, adipocytes exhibit particularly elevated levels of Cav-1, which constitutes approximately 30% of the total plasma membrane mass [157]. Several studies indicate that Cav-1 expression in adipose tissue is elevated in obese individuals. Fernández-Real et al. demonstrated that Cav-1 gene expression varies significantly between visceral and subcutaneous fat depots and is modulated by obesity status, suggesting a role for the CAV1 gene in lipid synthesis pathways, particularly in the context of obesity [124,158]. A comprehensive study involving human patients, in vitro human adipocytes, and mice further revealed that increased Cav-1 expression in adipose tissue from obese individuals correlates with elevated levels of inflammatory markers, including tumor necrosis factor-alpha (TNF-α) and nuclear factor kappa-light-chain-enhancer of activated B cells (NF-κB) [124]. This suggests a potential link between caveolae dysfunction and obesity-related metabolic disorders. Grayson et al. investigated the relationship between caveolae density, Cav-1 expression, and vascular function in an animal model of diet-induced obesity. Their findings revealed an increase in caveolae density and Cav-1 expression in both endothelial cells and smooth muscle cells of obese animals, which was associated with impaired nitric oxide (NO)-mediated vasodilation. These results suggest that structural and functional alterations in caveolae may contribute to vascular dysfunction in obesity [159]. Additionally, Razani et al. demonstrated that Cav-1-deficient mice exhibit reduced body weight and resistance to obesity, even when maintained on a high-fat diet for several months. The absence of Cav-1 was associated with the loss of membrane caveolae in adipocytes, a reduction in the female mammary gland fat pad, and a diminished hypodermal fat layer [153]. Collectively, these studies underscore the critical role of caveolae and caveolin proteins in adipose tissue biology and their involvement in obesity-related metabolic disorders. The caveolar network thus emerges as a promising therapeutic target for obesity and its associated complications.

## 4. Caveolae and Caveolin Proteins in Endothelial Dysfunction and Cardiovascular Disease

The vascular endothelium, a monolayer of squamous endothelial cells (ECs) lining blood vessels, is crucial for maintaining vascular homeostasis, regulating vascular tone, and modulating inflammatory and thrombotic responses. Under physiological conditions, ECs balance vasodilation and vasoconstriction, exhibit anti-inflammatory properties, and sustain a non-thrombogenic surface. However, exposure to risk factors such as hypertension, hyperlipidemia (e.g., elevated cholesterol and triglycerides), smoking, and diabetes can induce endothelial dysfunction, a key pathological event in the onset and progression of cardiovascular disease (CVD) [160,161,162]. CVD remains the leading cause of morbidity and mortality worldwide, posing a significant public health challenge. Despite advancements in diagnostics and therapeutics, a deeper understanding of the mechanisms driving CVD progression is essential for improving patient outcomes. In this context, the endothelium functions as a critical regulator of cardiovascular health, with ECs playing roles beyond their structural function as a barrier between circulating blood and adjacent tissues. Their remarkable plasticity across various organs underscores their potential as therapeutic targets in CVD [163,164,165]. Caveolae, specialized membrane invaginations abundant in vascular ECs, serve as essential signaling hubs, regulating endothelial function by sequestering key signaling molecules, including G protein-coupled receptors, receptor tyrosine kinases, and eNOS. This function is largely attributed to Cav-1, as evidenced by studies demonstrating that Cav-1 knockout not only abolishes caveolae, but also leads to pronounced endothelial dysfunction, characterized by dysregulated nitric oxide (NO) signaling and increased vascular permeability [25,88].

### 4.1. Caveolae and Caveolins in the Regulation of NO Synthesis in ECs

NO, synthesized by eNOS, is a crucial signaling molecule that mediates key EC-regulated processes [166,167,168]. In the vasculature, it modulates vascular tone and inhibits both platelet and leukocyte adhesion [169,170]. eNOS activity is tightly controlled by subcellular localization, protein–protein interactions and post-translational modifications [171]. Caveolae and their primary structural protein Cav-1 play a central role in eNOS regulation. Cav-1 directly binds eNOS via its scaffolding domain, sequestering the enzyme in an inactive state within caveolae. eNOS localization to caveolae is facilitated by myristoylation and palmitoylation, two post-translational modifications that promote its interaction with Cav-1 [172]. Genetic ablation of Cav-1 enhances NO synthesis and vasodilation, whereas Cav-1 overexpression suppresses eNOS activity, reducing NO bioavailability. Thus, aberrant Cav-1 expression and eNOS dysfunction, particularly eNOS uncoupling, which generates superoxide (O_2_^−^) instead of NO, are proposed as key contributors to endothelial dysfunction in cardiovascular disease (CVD).

### 4.2. The Role of Caveolae and Caveolins in Atherosclerosis

Atherosclerosis is a chronic inflammatory disease of large- and medium-sized arteries, characterized by lipid accumulation (dyslipidemia), immune cell infiltration, and extracellular matrix deposition within the vessel wall. These processes lead to plaque formation, resulting in vascular narrowing and stiffening [173,174]. It remains a leading cause of mortality in developed countries [175]. The disease progresses through multiple stages, with endothelial cells (ECs) playing a central role. The initial event involves endothelial dysfunction, often triggered by hypertension, smoking, hypercholesterolemia, and diabetes. This dysfunction increases vascular permeability and upregulates leukocyte adhesion molecules, facilitating the infiltration of low-density lipoprotein (LDL) and inflammatory cells into the vessel wall [173]. Once inside, LDL undergoes oxidation and is engulfed by macrophages, forming foam cells. The accumulation of foam cells gives rise to fatty streaks, the earliest detectable atherosclerotic lesions. Foam cell-derived cytokines and growth factors promote the migration and proliferation of smooth muscle cells from the tunica media into the tunica intima, where they contribute to fibrous plaque formation by producing extracellular matrix components. Over time, plaque destabilization and rupture can expose the thrombogenic lipid core to the bloodstream, triggering platelet aggregation and thrombus formation, which may result in acute vascular events such as myocardial infarction or stroke. Cav-1 has been implicated in critical stages of atherosclerotic plaque development, including endothelial dysfunction, lipid accumulation, and inflammation [176,177]. Given its role in LDL uptake, caveolae are potential targets for therapeutic intervention in atherogenesis [121]. Evidence suggests that reducing Cav-1 expression or genetically depleting Cav-1 confers atheroprotective effects. Le Master et al. demonstrated that Cav-1 loss inhibits oxidized LDL (ox-LDL)-induced endothelial stiffening, emphasizing its role in endothelial dysfunction associated with dyslipidemia [178]. Additionally, genetic deletion of Cav-1 in apolipoprotein E (ApoE)-deficient mice significantly reduces atherosclerotic plaque formation by decreasing LDL infiltration, leukocyte adhesion molecule expression, and monocyte recruitment to the vessel wall. Conversely, reintroducing Cav-1 specifically into ECs reverses these protective effects, highlighting the essential role of endothelial Cav-1 in atherogenesis [179,180]. Cav-1 also influences monocyte-to-macrophage differentiation via the early growth response 1 (EGR-1) pathway, suggesting a potential therapeutic target in atherogenesis [181]. Similarly, Tsai et al. reported that silencing Cav-1 in macrophages reduces Toll-like receptor 4 (TLR4) expression, decreases phagocytic activity and impairs bacterial clearance [182]. Other studies corroborate Cav-1’s role in regulating macrophage-driven inflammation by inhibiting TLR4-mediated cytokine production [183].

Caveolae and caveolins modulate intracellular signaling pathways, particularly those governing inflammation [184], which is central to leukocyte recruitment and the progression of atherosclerotic lesion [185]. In ECs, Cav-1 deficiency exacerbates inflammatory responses and promotes atherosclerotic plaque accumulation in animal models. Engel et al. demonstrated that Cav-1 deficiency mitigates atherosclerosis by attenuating inflammation, reducing leukocyte infiltration, and decreasing plaque size, while fostering an anti-inflammatory regulatory T-cell response [186]. These effects are associated with diminished endothelial expression of pro-inflammatory molecules such as CCL-2/MCP-1 and VCAM-1. Collectively, these findings position Cav-1 as a crucial regulator of endothelial inflammation and lipid metabolism. While its role remains context-dependent, targeting Cav-1-related pathways may present novel strategies for preventing or slowing atherosclerosis.

## 5. The Involvement of the Caveolar Network in Autoimmune Diseases

Autoimmune diseases encompass a diverse array of disorders wherein the immune system erroneously targets the body’s own tissues, resulting in chronic inflammation, tissue damage, and impaired organ function [187]. These conditions arise from a failure in immune tolerance, whereby self-reactive immune cells are inadequately regulated [188]. The pathogenesis of autoimmune diseases involves a complex interplay of genetic predisposition, environmental triggers, and immune dysregulation, including anomalies in T-cell and B-cell function, cytokine imbalances, and alterations in immune checkpoints. Autoimmune diseases can affect virtually any organ or system, leading to a broad spectrum of clinical manifestations ranging from mild to life-threatening. Elucidating the underlying mechanisms is crucial for the development of targeted therapies that can modulate immune responses and prevent disease progression. Although the role of caveolae and caveolins in autoimmune diseases remains under investigation, emerging evidence underscores their involvement in immune modulation and disease progression, highlighting their potential as therapeutic targets. To date, most of the information linking autoimmune diseases to the caveolar network relates to rheumatoid arthritis (RA) and systemic lupus erythematosus.

RA is a systemic autoimmune disease characterized by chronic inflammation of the synovial joints, leading to progressive joint destruction [189]. Over the past two decades, treatment strategies for RA have significantly advanced [190]. In RA, Cav-1 has been shown to interact with CD26 in antigen-presenting cells, leading to the phosphorylation of Cav-1 and subsequent activation of NF-κB, a key transcription factor in inflammatory responses [191]. This interaction promotes the upregulation of CD86 on APCs, enhancing T-cell activation and contributing to the chronic inflammation characteristic of RA. Additionally, Cav-1 expression is elevated in the synovial tissue of RA patients, correlating with increased vascularization and synoviocyte proliferation. Recent research has implicated Cav-1 in the pathogenesis of RA, particularly in its role in angiogenesis [192]. The study demonstrated that heat shock protein 70 (HSP70), a pro-angiogenic factor, accumulates in lipid rafts and is secreted extracellularly via exosomes, promoting ECs proliferation, migration and tube formation. Notably, the interaction between Cav-1 and HSP70 plays a crucial role in modulating HSP70-mediated angiogenesis [193]. Furthermore, the study revealed that pharmacological inhibition of HSP70 release suppresses a key signaling pathway without altering Cav-1 expression, ultimately attenuating pathological angiogenesis in RA. These findings suggest that targeting Cav-1–HSP70 interactions could offer novel therapeutic strategies to mitigate RA-associated vascular dysfunction. Overall, these findings suggest that Cav-1 not only plays a role in the immune dysregulation seen in RA, but also represents a potential therapeutic target for modulating immune responses and alleviating disease symptoms.

Systemic Lupus Erythematosus (SLE) is a chronic inflammatory disease that primarily affects the skin, joints, kidneys, blood cells, and nervous system, although other organs may also be involved [194]. Emerging evidence suggests a significant association between caveolin protein expression and SLE pathogenesis. Notably, Cav-1 has been identified as a differentially expressed gene in the B-cell transcriptomes of SLE patients, implicating its role in dysregulated signaling pathways associated with the disease [195]. Notably, a more detailed assessment indicated that Cav-3 may have greater diagnostic value than Cav-1 [196]. A recent study investigated the involvement of caveolin proteins in SLE, identifying Cav-1 and Cav-3 as potential diagnostic biomarkers [196]. The analysis revealed significantly elevated serum levels of Cav-1 and Cav-3 in SLE patients compared to healthy controls, while Cav-2 remained undetectable in both groups. Despite these increased levels, no significant correlation was observed between Cav-1 and Cav-3 concentrations and disease activity, suggesting that their role in SLE may be independent of disease severity. These findings highlight the potential of circulating caveolins as novel biomarkers for SLE and underscore the need for further research to elucidate their functional role in disease pathogenesis.

## 6. The Caveolar Network in Ocular Diseases

The human eye is a highly specialized organ responsible for vision, integrating multiple anatomical and functional components to process visual information (Figure 2). Its outermost layer comprises the cornea, a transparent avascular structure that provides most of the eye’s refractive power, and the sclera, which maintains the ocular shape. The anterior chamber contains aqueous humor, a fluid essential for ocular nourishment and intraocular pressure (IOP) homeostasis [197]. Most aqueous humor drains through the trabecular meshwork (TM), a spongy permeable structure at the iridocorneal angle, into Schlemm’s canal (SC), an endothelial-lined channel that transports fluid into the episcleral veins. Light refraction is fine-tuned by the biconvex lens, directing it onto the retina, where photoreceptors (rods and cones) convert light into neural signals [198]. These signals propagate through bipolar and ganglion cells to the optic nerve, transmitting visual data to the brain. The choroid, a vascularized layer, supplies oxygen and nutrients to the outer retina, while the retinal pigment epithelium (RPE) is essential for photoreceptor maintenance. The macula, particularly the fovea, is specialized for high-acuity vision due to its dense concentration of cone photoreceptors [199]. Impaired aqueous humor drainage can elevate IOP, increasing the risk of glaucoma, a group of disorders characterized by progressive retinal ganglion cell (RGC) degeneration and optic nerve damage, ultimately leading to irreversible blindness [200]. Among its forms, primary open-angle glaucoma (POAG) is the most prevalent [201].

Recent studies underscore the role of caveolae and caveolins in ocular pathophysiology, particularly in glaucoma [120,202]. Cav-1 is expressed in multiple ocular tissues, including the cornea, lens, retina, and TM, where it is crucial for maintaining ocular homeostasis, contributing to IOP regulation, retinal integrity, and corneal transparency [203]. In addition, caveolae function as mechanosensitive structures, playing a key role in detecting and modulating IOP fluctuations [203]. Research indicates that Cav-1 deficiency leads to elevated IOP, a condition reversible upon Cav-1 reintroduction into the TM [204,205]. Additionally, Cav-1 modulates RhoA signaling in the TM, influencing IOP regulation and POAG susceptibility [204]. Cav-1 is also implicated in IOP regulation via nitric oxide (NO) signaling, specifically by modulating NO bioavailability in the TM and SC. Lei et al. demonstrated that Cav-1 deficiency significantly enhances eNOS activity in ocular tissues, increasing NO production and leading to vascular abnormalities in retinal and choroidal blood vessels [206]. These findings highlight the critical role of Cav-1 in regulating eNOS activity to preserve normal ocular vascular function, suggesting that its dysregulation may contribute to ocular vascular pathologies. In a separate study, researchers assessed IOP and aqueous humor outflow in models with eNOS and Cav-1 gene deletions, revealing that eNOS deletion mitigates the adverse effects of Cav-1 deficiency on ocular fluid dynamics, reinforcing the importance of NO in IOP regulation [207]. Furthermore, Zhang et al. investigated the neuroprotective role of Cav-1 in glaucomatous RGCs, demonstrating that Cav-1 modulates retinal microglial polarization and inhibits inflammatory pathways. By downregulating Toll-like receptor 4 (TLR4) and activating the Akt/PTEN signaling pathway, Cav-1 reduces RGC apoptosis, underscoring its potential as a novel therapeutic target for glaucoma [208].

A recent study by Shui provides further insights into the role of caveolin proteins in glaucoma, with a particular focus on TM function and IOP regulation [209], highlighting the critical involvement of caveolae in facilitating aqueous humor outflow. Using RNA sequencing and qPCR analyses, the authors identified a significant reduction in cavin-2 expression in the TM of patients with primary POAG, particularly among individuals of African ancestry. Furthermore, transmission electron microscopy analysis revealed a disruption of caveolae ultrastructure in glaucomatous TM compared to healthy donors. These findings suggest that caveolin-associated mechanisms contribute to TM dysfunction and IOP dysregulation in glaucoma. The study also underscores the essential role of SDPR in maintaining TM integrity and highlights the potential for caveolar-targeted therapies in glaucoma treatment.

Diabetic retinopathy (DR) is a microvascular complication of diabetes mellitus, characterized by progressive retinal vascular damage that can lead to vision impairment and blindness. Its pathophysiology is complex, involving both vascular and neuronal dysfunction [210,211]. Caveolae and caveolins, particularly Cav-1, have been implicated in DR pathogenesis. Cav-1 plays a crucial role in retinal function, as its absence disrupts neurovascular coupling and impairs retinal responses [212]. Moreover, elevated Cav-1 expression has been detected in the vitreous and fibrovascular membranes of patients with proliferative DR, suggesting its involvement in disease progression [213]. Gu et al. investigated Cav-1’s role in maintaining blood–retinal barrier (BRB) integrity and vascular homeostasis. In a diabetic rat model, increased caveolar size and transcytosis were observed in the retina, indicating that Cav-1 dysregulation may contribute to DR development [213]. Their findings suggest that Cav-1 is critical for BRB integrity, venous stability, and mural cell homeostasis, underscoring its potential as a therapeutic target for retinal vascular diseases [214]. Hillman et al. examined the interplay between hypertension and diabetes in retinal capillary ultrastructure, demonstrating their synergistic effects in exacerbating microvascular changes. These included increased caveolae formation, reduced pericyte–endothelial interactions, and basement membrane thickening [215]. A recent study by Sasamoto has underscored the critical role of Cav-1 and Cav-2 in corneal regeneration by enhancing the proliferative capacity of basal cell adhesion molecule (BCAM)-positive corneal progenitors [216]. The findings demonstrate that both proteins are preferentially expressed in the basal epithelial layer of the cornea and limbus, where they regulate the surface expression of fibroblast growth factor receptor 2 (FGFR2), a key determinant of corneal epithelial proliferation and differentiation. Moreover, functional knockdown of both proteins significantly impairs the proliferative capacity of corneal progenitor cells, emphasizing their essential role in maintaining a regenerative phenotype. Beyond supporting the pivotal regulatory function of Cav-1 and Cav-2 in corneal progenitor cell dynamics, this study suggests that these proteins may serve as promising therapeutic targets for enhancing corneal regeneration.

Collectively, these studies highlight the pivotal role of caveolae and caveolins in DR, reinforcing their relevance as potential therapeutic targets.

## 7. Caveolae and Caveolins in the Central Nervous System (CNS)

Caveolae and caveolins are essential for maintaining CNS homeostasis by regulating neurotrophic signaling, blood–brain barrier (BBB) function, and lipid metabolism [217]. Early studies suggested that neurons lacked caveolae and caveolins [218], but subsequent research has confirmed that both neurons and glial cells, including astrocytes, express all caveolin isoforms, with Cav-1 being the most extensively studied in the CNS [219,220,221,222]. Additionally, Cav-1 phosphorylation is crucial for promoting axonal growth during early neuronal differentiation, indicating that modulation of this pathway may have therapeutic potential for neurodevelopmental disorders [223]. Cav-1 is predominantly expressed in ECs of the BBB, where it is thought to regulate permeability, potentially contributing to neuroinflammation and cognitive deficits [224,225,226]. Given the dysregulated expression and function of caveolins in neurodegenerative diseases such as Alzheimer’s, Parkinson’s, and Huntington’s, further understanding their role is vital for the development of targeted therapeutic strategies [217].

### 7.1. Caveolae and Caveolins in Alzheimer’s Disease (AD)

A hallmark of Alzheimer’s disease (AD) is the dysregulated metabolism of amyloid beta (Aβ), a peptide produced through the cleavage of amyloid precursor protein (APP) by enzymes, including β-secretases [227,228]. The accumulation of Aβ leads to plaque formation in the brain, disrupting neuronal communication, promoting inflammation, and driving neurodegeneration. Although the exact mechanisms underlying Aβ accumulation remain unclear, it is hypothesized that Aβ may misfold or aggregate into toxic forms, damaging neuronal structures and impairing cognitive function. Several studies have investigated the role of caveolins, particularly Cav-1, in AD, suggesting that Cav-1 expression may have both beneficial and detrimental effects on disease pathogenesis. Age-related changes in Cav-1 expression appear to play a significant role in the mechanisms underlying neurodegenerative diseases. Kang et al. [229] demonstrated that the upregulation of Cav-1 in aging neurons increases Aβ production by promoting APP cleavage via the β-secretase pathway, which may contribute to the development of sporadic Alzheimer’s disease. Further research by Gaudreault et al. [230] examined the relationship between Cav-1 expression and cholesterol homeostasis dysregulation in AD. Their analysis found elevated levels of Cav-1 protein and mRNA in post-mortem brain tissues from AD patients. Additionally, increased levels of Cav-3 have been shown to enhance β-secretase activity, promoting the cleavage of APP and the subsequent formation of Aβ peptides. The study indicated that Cav-3 upregulation facilitates the interaction between β-secretase and APP within caveolae, thus enhancing the cleavage process [231]. Scientific evidence also links reduced Cav-1 expression with the onset of AD. Bonds and colleagues [198] explored the relationship between type 2 diabetes mellitus (T2DM) and AD, finding reduced Cav-1 levels in both the brains of T2DM patients and in diabetic mouse models. This reduction was associated with elevated levels of Aβ and hyperphosphorylated tau (p-tau), key markers of AD pathology. Notably, restoring Cav-1 expression in diabetic mice improved cognitive function and alleviated AD-related pathology. In a separate study, Islam [232] observed that the absence of Cav-1 during fetal development caused metabolic abnormalities, disrupting normal brain aging. This highlights the critical role of Cav-1 in establishing proper metabolic programming during fetal brain development. Therefore, modulating Cav-1 or Cav-3 expression or activity may represent a promising therapeutic strategy for the prevention or treatment of AD.

### 7.2. Involvement of Caveolins in Parkinson’s Disease (PD)

PD is a neurodegenerative disorder primarily characterized by motor deficits resulting from the progressive loss of dopamine-producing neurons, particularly in the substantia nigra. This neuronal degeneration leads to a marked reduction in dopamine levels, especially in brain regions responsible for motor control. Additionally, PD is defined by the abnormal accumulation of alpha-synuclein, a protein crucial for various physiological processes in the brain. Alpha-synuclein interacts with lipid membranes, facilitates synaptic vesicle clustering and recycling, regulates dopamine release and modulates reactive oxygen species (ROS) homeostasis [233]. However, in PD, the misfolding and aggregation of alpha-synuclein contribute to neuronal dysfunction and death. Other pathological features of PD include chronic microglial inflammation, mitochondrial dysfunction impairing cellular energy production, and increased oxidative stress [234,235]. While PD is primarily sporadic, genetic mutations, particularly in the PARK2 gene that encodes the E3 ubiquitin ligase, have been linked to its pathogenesis. A study by Cha identified Cav-1 as a novel substrate of Parkin, providing insight into how parkin deficiency may contribute to PD pathology. Cav-1 accumulation disrupts lipid raft function and enhances α-synuclein transmission to neighboring cells, potentially accelerating disease progression [236]. Cav-1 expression increases with aging, facilitating α-synuclein intercellular transmission and accumulation [237]. Studies further suggest that the direct interaction between Cav-1 and α-synuclein contributes to its toxicity. Moreover, RNA interference-mediated downregulation of Cav-1 mitigates α-synuclein-induced cell death, indicating that Cav-1 may serve as a promising therapeutic target to reduce α-synuclein toxicity in PD [237].

### 7.3. The Role of Caveolins in the Nervous System Tumors

Brain tumors represent a heterogeneous group of neoplasms originating from various brain tissues, each exhibiting unique clinical characteristics. Glioblastomas, in particular, account for approximately half of all malignant brain tumors [238,239]. Among the different members of the caveolin family of proteins, Cav-1 has been extensively studied for its dual role in cancer biology, acting as either a tumor suppressor or promoter, contingent upon the tumor context [240]. In the realm of neuro-oncology, emerging evidence underscores the significance of caveolae and caveolins, particularly Cav-1 in the pathogenesis of brain tumors. Cav-1 is a key biomarker and potential therapeutic target in glioblastoma, as its high expression correlates with tumor aggressiveness and poor patient outcomes. A recent analysis of clinical datasets revealed a strong correlation between high Cav-1 levels and poor prognosis, particularly in the proneural and mesenchymal glioblastoma subtypes [241]. Increased Cav-1 expression was also associated with IDH wild-type status, higher histological tumor grade, and lower Karnofsky Performance Score. Pathway analysis further implicated Cav-1 in epithelial-to-mesenchymal transition, cell adhesion, and extracellular matrix remodeling, reinforcing its role in tumor invasiveness. Additionally, the study identified a subset of 27 genes co-expressed with Cav-1, among which matrix metalloproteinases (MMPs) 2 and 9 and PAI-1 demonstrated synergistic prognostic value when combined with Cav-1 [241]. A recent study found that Cav-1 plays a crucial role in the PI3K/Akt-dependent upregulation of plasminogen activator inhibitor-1 (PAI-1), which promotes glioma proliferation and metastasis through an enhanced epithelial–mesenchymal transition (EMT) and angiogenesis. Consequently, Cav-1 knockdown significantly inhibits glioma proliferation and metastasis [242]. Pu et al. demonstrated that both Cav-1 and cavin-1 are upregulated in glioblastoma compared to normal tissues, with their expression correlating with increased levels of urokinase plasminogen activator and gelatinases [243]. These findings further support the hypothesis that modulating Cav-1 or cavin-1 expression in glioblastoma cells affects tumor invasiveness. A recent study explored the role of Sprouty (SPRY) proteins—key regulators of receptor tyrosine kinase (RTK) signaling—in modulating FGFR1 endocytosis and degradation in glioblastoma. Notably, FGFR1 expression is typically elevated in glioblastoma relative to normal brain tissue and has been associated with resistance to radiotherapy [244]. Cav-1 was found to regulate SPRY2 overexpression, which reduces Cav-1 vesicles, thereby inhibiting FGFR1 signaling while enhancing EGFR activation [245]. These molecular interactions may influence glioblastoma cell behavior and their response to therapy. Kagawa and colleagues have demonstrated that fatty acid-binding protein 7 (FABP7) is highly expressed in the nuclei of Isocitrate dehydrogenase 1 (IDH1) wild-type glioblastoma and promotes tumor proliferation by upregulating Cav-1 expression [246]. Specifically, FABP7 enhances caveolae/caveosome formation by inducing histone acetylation at the Cav-1 promoter, but only in IDH1 wild-type glioblastomas. In contrast, IDH1-mutant glioblastomas exhibit lower Cav-1 expression, reduced histone acetylation and decreased acetyl-CoA levels. Supporting these findings, the authors further showed that mutant FABP7 disrupts these effects, underscoring its role in tumor aggressiveness [246]. Beyond glioblastoma, the role of caveolins extends to other brain tumors, including oligodendrogliomas and astrocytomas and neuroblastoma. A study by Senetta identified Cav-1 expression as an independent prognostic marker in oligodendroglial tumors, correlating with higher tumor grades and reduced patient survival [247]. Immunohistochemical analysis revealed that Cav-1 expression was predominantly detected in high-grade mixed oligoastrocytomas and glioblastomas with an oligodendroglial component, whereas low-grade tumors exhibited minimal expression. Furthermore, Cav-1 positivity was significantly associated with the absence of 1p/19q codeletion, a well-established marker of favorable prognosis in oligodendrogliomas. Overall, these findings indicate that Cav-1 acts as an oncogene in gliomas, promoting tumor progression and contributing to poor prognosis. A recent study by Riitano and colleagues underscores the pivotal role of lipid rafts in neuroblastoma, a type of brain tumor originating from glial cells. The authors demonstrate that LRP6, a critical co-receptor in the oncogenic Wnt/β-catenin signaling pathway, is localized to these lipid microdomains following tissue plasminogen activator (tPA) stimulation. Furthermore, their findings indicate that disruption of lipid rafts impairs LRP6 phosphorylation and β-catenin signaling. These results suggest that targeting lipid rafts could offer a promising therapeutic approach to modulating LRP6 activity in neuroblastoma cells [248]. Therefore, while further research is needed to fully elucidate the biological functions and therapeutic potential of the caveolar network, targeting Cav-1 and caveolae-associated pathways holds promise as a novel therapeutic strategy for gliomas and other brain tumors.

## 8. Conclusions and Prospective Therapeutic Strategies Targeting Caveolae/Caveolins in Human Disease

The growing recognition of caveolae and caveolin proteins as key regulators in human disease has unveiled promising avenues for therapeutic intervention that targets these specialized membrane microdomains. Given their fundamental roles in cellular signaling, endocytosis, and mechanotransduction, caveolins have emerged as potential pharmacological targets across a broad spectrum of disorders, including metabolic diseases, cardiovascular conditions, ocular pathologies, nervous system tumors, and autoimmune disorders. One particularly compelling strategy involves caveolin-derived peptides, such as Cavtratin, a Caveolin-1 scaffolding domain peptide, which has demonstrated potential in mitigating endothelial dysfunction and attenuating inflammation in cardiovascular and metabolic diseases. Additionally, nanotechnology-based approaches are being explored to optimize drug delivery to caveolae-rich tissues, enhancing therapeutic precision while minimizing off-target effects. In oncology, lipid raft modulators and cholesterol-depleting agents, such as MβCD, have been investigated for their ability to disrupt caveolae-associated oncogenic signaling in glioblastoma and other nervous system malignancies, although systemic toxicity remains a concern. In ophthalmology, caveolin modulation has been proposed as a strategy for regulating intraocular pressure in glaucoma and promoting epithelial regeneration in corneal degeneration. Furthermore, in autoimmune diseases such as RA and systemic lupus erythematosus, targeting caveolae-mediated immune signaling may offer novel approaches for restoring immune homeostasis. Future research should prioritize refining these therapeutic strategies, identifying disease-specific caveolin modulators and advancing personalized caveolae-targeted treatments with improved specificity and safety. The integration of gene therapy, nanomedicine, and synthetic caveolin mimetics may represent the next frontier in precision medicine, unlocking novel opportunities for the treatment of caveolae-related diseases.

## Figures and Tables

**Figure 1 ijms-26-02918-f001:**
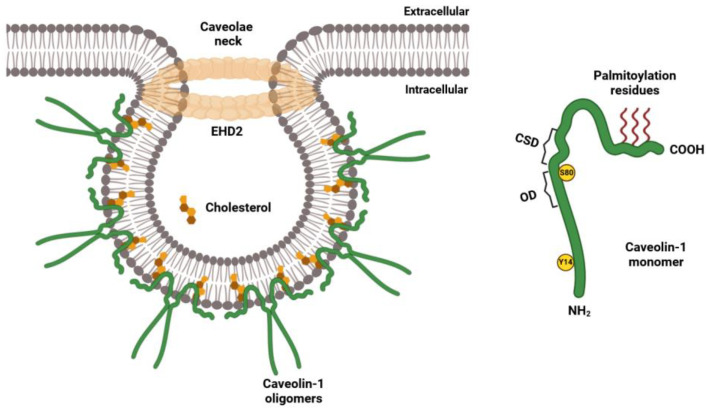
Model of caveolae and Cav-1 domains. CSD, caveolin scaffolding domain; EHD2, Eps15 homology domain 2; OD, oligomerization domain; S80, phosphorylation at Serine 80; 14, phosphorylation at Tyrosine Y14. (Created with Biorender.com).

**Figure 2 ijms-26-02918-f002:**
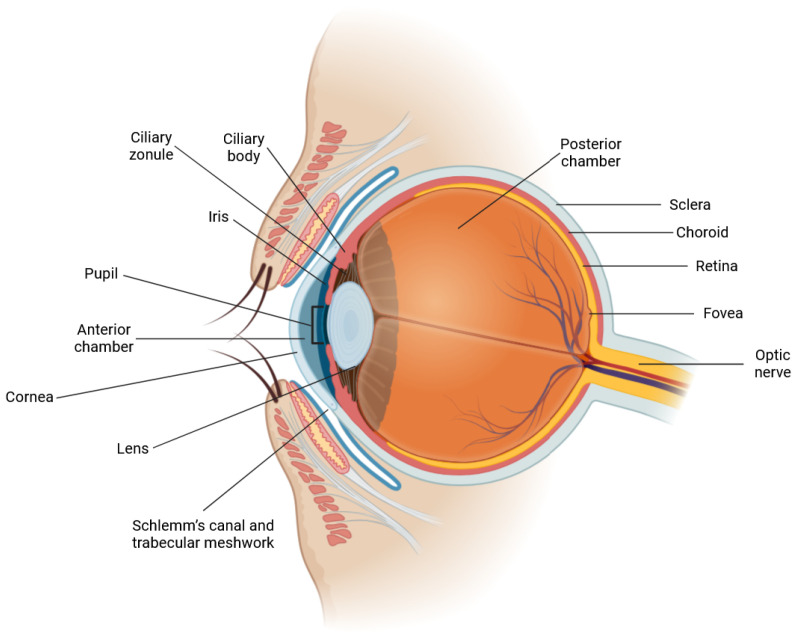
Anatomy of the eye (created with Biorender.com).

**Table 1 ijms-26-02918-t001:** Types and main functions of lipids found in lipid rafts.

Lipid Type	Contribution to Lipid Rafts
Cholesterol	Maintains fluidity and structural integrity; critical for raft stability.
Gangliosides	Facilitate signaling by interacting with proteins and lipids.
Glycosphingolipids	Contribute to raft stability and specific protein recruitment.
Saturated Phospholipids	Enhance packing density and lipid raft rigidity.
Sphingolipids	Provide more ordered packing; interact tightly with cholesterol.

**Table 2 ijms-26-02918-t002:** Structural and functional comparison of lipid rafts and caveolae.

	Lipid Rafts	Caveolae	References
Definition	Microdomains of the plasma membrane rich in cholesterol, sphingolipids, and certain proteins.	Flask-shaped invaginations of the plasma membrane enriched in cholesterol, sphingolipids, and caveolins.	[5,85]
Shape and Structure	Non-distinct, flat, or slightly curved regions of the membrane.	Flask- or omega-shaped membrane invaginations.	[35,86]
Size	10–200 nm.	Typically, 50–100 nm in diameter.	[87]
Protein Markers	Glycosylphosphatidylinositol (GPI)-anchored proteins, flotillins.	Cav-1, -2, -3, and Cavins.	[88,89]
Function	Signal transduction, membrane sorting, and lipid/protein trafficking.	Signal transduction, endocytosis, mechanosensing, and cholesterol homeostasis.	[5,90]
Dependence on Cholesterol	Cholesterol is essential for maintaining raft integrity and functionality.	Heavily reliant on cholesterol for structural stability and caveolin-membrane interaction.	[91]
Biogenesis	Formed dynamically through lipid–lipid and lipid–protein interactions.	Formed through the interaction of caveolins with the membrane lipids and cytoskeletal elements.	[35]
Presence in Cells	Ubiquitous in eukaryotic cells.	Found predominantly in adipocytes, endothelial cells, and muscle cells.	[89]

**Table 3 ijms-26-02918-t003:** Commonly reported drugs and chemicals with potential to affect lipid rafts/caveolae and their mechanisms of action.

Drug/Chemical	Mechanism of Action	Main References
Brefeldin A	Inhibits Golgi function and vesicular transport, preventing caveolae recycling	[81,92]
Cav-based cell permeable peptides (e.g., Cavtratin)	Mimics Cav-1 function, inhibiting excessive signaling	[93,94]
Cholesterol oxidase	Oxidizes cholesterol, disrupting lipid raft structure	[95,96]
Daidzein	It has been shown to modulate the expression of Cav-1, thereby affecting the biogenesis and function of caveolae	[97,98,99]
Dynamin inhibitors (e.g., dynasore)	Block caveolae-mediated endocytosis by inhibiting dynamin	[100,101,102]
Filipin	Binds to cholesterol, disrupting lipid raft integrity	[103,104]
Lovastatin	Inhibits HMG-CoA reductase, reducing cholesterol synthesis	[105,106]
MβCD	Removes cholesterol from lipid rafts, disrupting their structure	[50,107]
Nystatin	Binds to cholesterol, disrupting lipid raft structure	[103,108]

**Table 4 ijms-26-02918-t004:** Metabolic disorders and their links to Cav-1 dysfunction.

Metabolic Disorder	Pathophysiology	Link to Cav-1 Dysfunction	References
Cardiovascular Diseases	Dysregulated cholesterol metabolism and endothelial dysfunction.	Cav-1 mutations lead to cholesterol efflux defects and reduced nitric oxide bioavailability, promoting vascular diseases.	[56]
Diabetes Mellitus	Impaired glucose metabolism and insulin resistance.	Dysregulation of Cav-1 affects insulin receptor signaling and GLUT4 translocation, exacerbating insulin resistance.	[119,120]
Dyslipidemia	Abnormal lipid levels, including elevated LDL and triglycerides.	Altered caveolar lipid homeostasis due to Cav-1 dysregulation affects lipid uptake and efflux in hepatocytes and adipocytes.	[121]
Lipodystrophy	Abnormal fat distribution and metabolic derangements.	Loss of caveolae in adipocytes due to Cav-1 mutations cause lipodystrophic phenotypes with insulin resistance.	[122]
Non-Alcoholic Fatty Liver Disease (NAFLD)	Hepatic lipid accumulation and steatosis.	Impaired caveolar lipid metabolism influences hepatic lipid storage and triglyceride secretion, driving NAFLD progression.	[123]
Obesity	Excess adipose tissue accumulation and systemic inflammation.	Cav-1 dysfunction disrupts adipocyte differentiation and lipid storage, contributing to adipose tissue dysfunction.	[123,124]

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
