# Peer review of "Caveolae: Metabolic Platforms at the Crossroads of Health and Disease"

_ijms, 2025, doi:10.3390/ijms26072918_

Round 1
Reviewer 1 Report
Comments and Suggestions for Authors
The article of Dante Maria Stea and Alessio D’Alessio explores in deep caveolae and caveolin proteins, emphasizing their roles in metabolism, lipid regulation, insulin signaling, and cardiovascular function.
Although the manuscript is well-structured I have a few requests:
- I would suggest adding two short paragraphs on the importance of caveolae and/or lipid rafts in nervous system tumors and autoimmune diseases. This addition would provide a more comprehensive view of how lipid rafts influence disease mechanisms, highlighting their role in signal transduction pathways.
-LRP6 is only briefly mentioned, but it plays a significant role in various diseases, including tumors such as neuroblastoma. Indeed, a study demonstrates that the signal transduction pathway mediated by LRP6, activated by tissue plasminogen activator and inhibited by methyl-β-cyclodextrin (MβCD), functions through lipid rafts in neuroblastoma cells, suggesting a crucial role for these microdomains in signal transduction. I would like the authors add this reference.
-A short table about drugs which interact with lipid rafts should be added
Author Response
The article of Dante Maria Stea and Alessio D’Alessio explores in deep caveolae and caveolin proteins, emphasizing their roles in metabolism, lipid regulation, insulin signaling, and cardiovascular function.
Although the manuscript is well-structured I have a few requests:
- I would suggest adding two short paragraphs on the importance of caveolae and/or lipid rafts in nervous system tumors and autoimmune diseases. This addition would provide a more comprehensive view of how lipid rafts influence disease mechanisms, highlighting their role in signal transduction pathways.
Authors’ reply:
We thank the reviewer for her/his suggestion. We have added two new paragraphs, titled "5. The involvement of the caveolar network in autoimmune diseases" and the sub-paragraph "6.3 The role of caveolins in nervous system tumors," highlighted in red for better readability. We hope this addresses the reviewer's comment.
- LRP6 is only briefly mentioned, but it plays a significant role in various diseases, including tumors such as neuroblastoma. Indeed, a study demonstrates that the signal transduction pathway mediated by LRP6, activated by tissue plasminogen activator and inhibited by methyl-β-cyclodextrin (MβCD), functions through lipid rafts in neuroblastoma cells, suggesting a crucial role for these microdomains in signal transduction. I would like the authors add this reference.
Authors’ reply:
We thank the reviewer for her/his valuable suggestion. We have accordingly added a few lines (782-789) describing the crucial role that LRP6 plays as a critical co-receptor in the oncogenic Wnt/β-catenin signaling pathway in neuroblastoma. We hope that we have identified the correct reference to which the reviewer is referring.
- A short table about drugs which interact with lipid rafts should be added
Authors’ reply:
Following the reviewer's suggestion, we have included a new Table 2 (page 6, highlighted in red) summarizing the most common drugs/chemicals with potential to affect the biogenesis and functions of lipid rafts and caveolae. We have also included information on the reported mechanism of action for each of them.
Reviewer 2 Report
Comments and Suggestions for Authors
This review article (manuscript ID #: ijms-3528357) first introduced and compared lipid rafts, caveolae, and caveolins, then summarized the correlation of caveolae/caveolins with cell metabolism and metabolic disorders (e.g., glucose metabolism, glycolysis, diabetes, obesity, endothelial dysfunction, etc.), and subsequently reviewed the role of Caveolae/caveolins in various diseases including atherosclerosis, ocular diseases, Alzheimer’s disease, and Parkinson’s disease. The manuscript is organized well, and the language is pretty good. I have the following comments.
- The research on lipid rafts and caveolae has lasted for ~30 years, and a lot of review articles on all kinds of topics, including the role of caveolae in metabolic disorders, have been published. What new information/knowledge and latest advances does this review article provide? I do not get it.
- There are only 2 figures and 3 tables in this manuscript. The two figures and Table 1 provide no novel information or knowledge. Table 2 displays the structural and functional comparison of lipid rafts and caveolae. It just provides some well-known common information/knowledge (moreover, all references were published more than 10 years ago). Table 3 potentially provide a little new information/knowledge, however, it does not present the latest advances because most references were published more than 14 years ago and only two references (i.e., references 100 and 103) were published 5 years ago (i.e., in 2020).
- More figures and/or tables should be provided to summarize and display the latest advances in some sections or subsections.
- I am not sure whether it is appropriate to include sections 5 (ocular diseases) and 6 (CNS diseases) in this article.
- The abstract mentions “… discusses potential therapeutic strategies targeting caveolin function and the stabilization of caveolae to restore metabolic homeostasis”. It would be better to particularly discuss the potential therapeutic strategies in an independent section before the Conclusions section.
- Line 164 in the legend to Figure 1: “14” should be “Y14”.
- “in vitro” (e.g., lines 70 and 71) and “in vivo” (e.g., 186) should be in italics. Please check it throughout the main text.
Author Response
This review article (manuscript ID #: ijms-3528357) first introduced and compared lipid rafts, caveolae, and caveolins, then summarized the correlation of caveolae/caveolins with cell metabolism and metabolic disorders (e.g., glucose metabolism, glycolysis, diabetes, obesity, endothelial dysfunction, etc.), and subsequently reviewed the role of Caveolae/caveolins in various diseases including atherosclerosis, ocular diseases, Alzheimer’s disease, and Parkinson’s disease. The manuscript is organized well, and the language is pretty good. I have the following comments.
- The research on lipid rafts and caveolae has lasted for ~30 years, and a lot of review articles on all kinds of topics, including the role of caveolae in metabolic disorders, have been published. What new information/knowledge and latest advances does this review article provide? I do not get it
Authors’ reply:
We appreciate the reviewer's comment regarding the extensive literature on the role of caveolae and their associated structural proteins in various cellular and molecular mechanisms, as well as their involvement in human diseases. In response, we have updated the references to incorporate the most recent studies available whenever possible. Furthermore, we have added a new paragraph, "Prospective Therapeutic Strategies Targeting Caveolae/Caveolin to Restore Cellular Functions," to provide an overview of the latest advancements and emerging insights in the field. We hope this will answer the referee's comment.
- There are only 2 figures and 3 tables in this manuscript. The two figures and Table 1 provide no novel information or knowledge. Table 2 displays the structural and functional comparison of lipid rafts and caveolae. It just provides some well-known common information/knowledge (moreover, all references were published more than 10 years ago). Table 3 potentially provides a little new information/knowledge, however, it does not present the latest advances because most references were published more than 14 years ago and only two references (i.e., references 100 and 103) were published 5 years ago (i.e., in 2020)
Authors’ reply:
We acknowledge the omission of additional graphical or tabular data in the previous version of the manuscript and appreciate the reader’s feedback. While we recognize that Table 1 and Figure 1 do not introduce novel information regarding the composition and organization of lipid rafts, caveolae, and the proteins involved in their biogenesis, we included them to enhance clarity, particularly for readers less familiar with the field. Additionally, we have updated the reference list, replacing some older citations with more recent literature. However, we have retained certain foundational articles that, despite their age, remain seminal contributions to this area of research.
- More figures and/or tables should be provided to summarize and display the latest advances in some sections or subsections
Authors’ reply:
Thank you for your valuable suggestion. In response, we have incorporated a new paragraph titled 'Prospective Therapeutic Strategies Targeting Caveolae/Caveolin to Restore Cellular Functions,' which highlights the latest advancements in the field. Additionally, we have added Table 2, which presents a selection of commonly used drugs and chemical compounds that modulate the functionality of caveolae and lipid rafts, along with relevant recent references.
- I am not sure whether it is appropriate to include sections 5 (ocular diseases) and 6 (CNS diseases) in this article
Authors’ reply:
We firmly believe that caveolae and their structural proteins play a crucial role in ocular pathophysiology, with their molecular mechanisms only now beginning to be elucidated. Accordingly, we have revised this section to incorporate recent findings that further support this perspective. Given its relevance, we would greatly appreciate retaining this discussion in the final version of the manuscript.
- The abstract mentions “… discusses potential therapeutic strategies targeting caveolin function and the stabilization of caveolae to restore metabolic homeostasis”. It would be better to particularly discuss the potential therapeutic strategies in an independent section before the Conclusions section
Authors’ reply:
We sincerely appreciate the reader's valuable suggestion. In response, we have thoroughly revised the final section of the manuscript and retitled it "Conclusions and Prospective Therapeutic Strategies Targeting Caveolae/Caveolin in Human Disease." We believe this revised section more effectively addresses the reviewer's comments and enhances the overall quality of our work.
- Line 164 in the legend to Figure 1: “14” should be “Y14”.
Authors’ reply:
Thank you for spotting this typo. We have amended the text as suggested. All corrections are indicated in red in the text.
- “in vitro” (e.g., lines 70 and 71) and “in vivo” (e.g., 186) should be in italics. Please check it throughout the main text.
Authors’ reply:
Thank you for noticing these typos. We have modified the text as suggested. All corrections are indicated in red in the text.
Reviewer 3 Report
Comments and Suggestions for Authors
The review by Dante Maria Stea and Alessio D’Alessio provides an in-depth analysis of caveolae and caveolin proteins, detailing their role in cellular metabolism, lipid homeostasis, insulin signaling, and cardiovascular regulation. The article effectively links caveolin dysfunction to metabolic disorders (obesity, type 2 diabetes), cardiovascular diseases, neurodegenerative disorders (Alzheimer’s, Parkinson’s), and ocular diseases (glaucoma, diabetic retinopathy).
The authors offer a well-structured discussion of caveolae as membrane microdomains essential for signal transduction and endocytosis, emphasizing their therapeutic potential in metabolic and vascular disorders. It is well written and provides a complete review of the literature in this field.
However, I would appreciate some further insights.
I believe the work would be better appreciated if a short paragraph on autoimmune diseases is included. In fact, recent authors have described molecular advancements related to lipid rafts.
Moreover, I would appreciate a more critical discussion on cholesterol-depleting agents or lipid rafts modulators.
Could you create a table of the main drugs that act on caveolae and/or lipid rafts?
Author Response
The review by Dante Maria Stea and Alessio D’Alessio provides an in-depth analysis of caveolae and caveolin proteins, detailing their role in cellular metabolism, lipid homeostasis, insulin signaling, and cardiovascular regulation. The article effectively links caveolin dysfunction to metabolic disorders (obesity, type 2 diabetes), cardiovascular diseases, neurodegenerative disorders (Alzheimer’s, Parkinson’s), and ocular diseases (glaucoma, diabetic retinopathy).
The authors offer a well-structured discussion of caveolae as membrane microdomains essential for signal transduction and endocytosis, emphasizing their therapeutic potential in metabolic and vascular disorders. It is well written and provides a complete review of the literature in this field.
However, I would appreciate some further insights.
- I believe the work would be better appreciated if a short paragraph on autoimmune diseases is included. In fact, recent authors have described molecular advancements related to lipid rafts.
Authors’ reply:
We appreciate the reviewer’s suggestion and have incorporated a new paragraph titled "5. The Involvement of the Caveolar Network in Autoimmune Diseases." To facilitate the identification of the new and revised sections, all changes have been highlighted in red throughout the manuscript.
- Moreover, I would appreciate a more critical discussion on cholesterol-depleting agents or lipid rafts modulators.
Authors’ reply:
This is an important point, particularly in the context of interpreting experimental results. We appreciate the reviewer’s suggestion and have accordingly added a brief comment on this issue on page 6, beginning at line 263. For clarity, the revised text has been highlighted in red.
- Could you create a table of the main drugs that act on caveolae and/or lipid rafts?
Authors’ reply:
In response to the reviewer’s suggestion, we have added a new Table 2, which lists commonly used drugs and chemical compounds that modulate the functionality of caveolae and lipid rafts, accompanied by relevant recent references.
Round 2
Reviewer 1 Report
Comments and Suggestions for Authors
The authors have satisfied my requests
Reviewer 2 Report
Comments and Suggestions for Authors
This version of manuscript updates some references and added some new contents and an additional table according to my previous comments. However, I do not think the revision sufficiently summarizes the latest advances and provides sufficient novel information/knowledge both in the main text and in figures/tables for publication in IJMS, and I do not think a further revision can realize it. By the way, only updating the references cannot make the old descriptions becoming novel knowledge or new advances.
Reviewer 3 Report
Comments and Suggestions for Authors I believe that the authors of the present manuscript (ijms-3528357) have improved it according to all the reviewer's requests and now it is suitable for publication.